# Infection Prevention and Control in Three Tertiary Healthcare Facilities in Freetown, Sierra Leone during the COVID-19 Pandemic: More Needs to Be Done!

**DOI:** 10.3390/ijerph19095275

**Published:** 2022-04-26

**Authors:** Ibrahim Franklyn Kamara, Sia Morenike Tengbe, Bobson Derrick Fofanah, James Edward Bunn, Charles Kuria Njuguna, Christiana Kallon, Ajay M. V. Kumar

**Affiliations:** 1World Health Organization, 21A-B Riverside, Off King Harman Road Freetown, Freetown 00232, Sierra Leone; fofanahb@who.int (B.D.F.); bunnj@who.int (J.E.B.); njugunach@who.int (C.K.N.); 2Ministry of Health and Sanitation, 4th Floor, Youyi Building, Brookfields, Freetown 00232, Sierra Leone; siamoreniketengbe@outlook.com (S.M.T.); christy.conteh@yahoo.com (C.K.); 3International Union Against Tuberculosis and Lung Disease, 2 Rue Jean Lantier, 75001 Paris, France; akumar@theunion.org; 4International Union Against Tuberculosis and Lung Disease, South-East Asia Office, C-6 Qutub Institutional Area, New Delhi 110016, India; 5Yenepoya Medical College, Yenepoya (Deemed to Be University), University Road, Deralakatte 575018, India

**Keywords:** SORT IT (Structured Operational Researh Training Initiative), operational research, Sierra Leone, Infection Prevention and Control, WHO IPCAF (Infection Prevention and Control Assessment Framework) tool, personal protective equipment, COVID-19

## Abstract

Infection Prevention and Control (IPC) measures are critical to the reduction in healthcare-associated infections, especially during pandemics, such as that of COVID-19. We conducted a hospital-based cross-sectional study in August 2021 at Connaught Hospital, Princess Christian Maternity Hospital and Ola During Children’s Hospital located in Freetown, Sierra Leone. We used the World Health Organization’s Infection Prevention and Control Assessment Framework Tool to assess the level of IPC compliance at these healthcare facilities. The overall IPC compliance score at Connaught Hospital was 323.5 of 800 points, 313.5 of 800 at Ola During Children’s Hospital, 281 of 800 at Princess Christian Maternity Hospital, implying a ‘Basic’ IPC compliance grade. These facilities had an IPC program, IPC committees and dedicated IPC focal persons. However, there were several challenges, including access to safe and clean water and insufficient quantities of face masks, examination gloves and aprons. Furthermore, there was no dedicated budget or no healthcare-associated infection (HAI) surveillance, and monitoring/audit of IPC practices were weak. These findings are of concern during the COVID-19 era, and there is an urgent need for both financial and technical support to address the gaps and challenges identified.

## 1. Introduction

Antimicrobial Resistance (AMR) is a silent pandemic and a global public health threat that the world is ill-prepared to respond to [1]. In Low- and Middle-Income Countries (LMICS) such as Sierra Leone, AMR is driven by inappropriate use of antimicrobial agents, indiscriminate use of antimicrobial agents as growth promoters and a lack or poor implementation of legislation to combat AMR [2]. 

The World Health Organization’s (WHO) global action plan to combat AMR has five strategic objectives, one of which is to reduce the incidence of infections through sanitation, hygiene and by instituting Infection Prevention and Control (IPC) measures [3,4]. Every infection prevented means one less instance of antimicrobial use and thus a lower chance of development of AMR. IPC measures are also important given the increasing incidence of healthcare-associated infections (HAI) and the increasing frequency of disease outbreaks, including the ongoing COVID-19 pandemic in sub-Saharan Africa [5,6].

The ongoing COVID-19 pandemic has had devastating global socioeconomic and health consequences. In response to the COVID-19 pandemic, healthcare facilities worldwide have stepped up their IPC efforts. IPC helps to reduce the spread of infections, including those caused by resistant microbes, and promotes health worker and patient safety. Importantly, good IPC standards also help to protect vulnerable frontline healthcare workers given the limited and effective treatment options against COVID-19 [7,8,9]. 

In Sierra Leone, the National Infection Prevention and Control Unit (NIPCU) was established in 2015 during the 2014–2016 Ebola outbreak to strengthen the country’s response to infectious disease outbreaks and provide a safe environment for patients, visitors and healthcare workers. NIPCU has been crucial in reducing the risk of COVID-19 transmission in healthcare facilities [10,11]. The key initiatives include training of healthcare workers, ensuring adequate supplies of IPC materials, quarterly supportive site visits and the implementation of standard and transmission-based precautions, such as hand hygiene practices, use of personal protective equipment and isolation of suspected COVID-19 patients [10].

Despite the establishment of NIPCU, data regarding the implementation of IPC programs in healthcare facilities in Sierra Leone are scarce. In 2019, with the support of WHO, the Ministry of Health of Sierra Leone (MOHS) conducted a baseline assessment of IPC standards in tertiary care hospitals using the Infection Prevention and Control Assessment Framework (IPCAF) tool. The average IPCAF assessment scores ranged from ‘Inadequate’ to ‘Basic’ grades of compliance, as detailed in Table 1. Another study conducted in the pre-COVID-19 era evaluating IPC compliance at regional hospitals and selected peripheral health units using an MOHS assessment tool showed that compliance increased from 69% in 2016 to 73% in 2018 (expected minimal threshold =70%; desired threshold ≥85%) [12]. 

A recent study conducted in three secondary hospitals in Sierra Leone reported a high rate (28.9%) of secondary infection among healthcare workers. The poor healthcare worker-to-patient ratio that is seen in all healthcare facilities in the country will increase the incidence of healthcare-associated infections such as COVID-19 among healthcare workers. Additionally, poor IPC programs at healthcare facilities will increase the risk of transmission of COVID-19 among healthcare workers [13]. Identifying strengths and gaps in the implementation of IPC programs during the COVID-19 pandemic will inform the policymakers on how to improve standard operating procedures and will help them devise strategies to reinforce IPC, which in turn should reduce COVID-19 transmission and prevent AMR. This is particularly relevant in tertiary care facilities, which severe COVID-19 patients are often referred to for advanced care. The findings from this study will add to the global and national body of evidence on the implementation of IPC programs at healthcare facilities. We therefore undertook this research to (i) assess IPC compliance at three tertiary healthcare facilities in Freetown, Sierra Leone, using the WHO Infection Prevention and Control Facility Assessment Framework (IPCAF) and (ii) report on strengths and gaps in various components of IPC implementation.

## 2. Methods

### 2.1. Study Design

This was a hospital-based cross-sectional study involving primary data collection.

### 2.2. Study Setting

#### 2.2.1. General Setting

Sierra Leone is bordered by Guinea, Liberia and the Atlantic Ocean, and is divided into 16 districts with an estimated population of 8 million people with most (59%) of the population living in rural areas [14]. In 2018, the life expectancy was 53 years for males and 55 years for females, with communicable diseases accounting for about 57% of all deaths. The total expenditure on health as a percentage of Gross Domestic Product was 16% [15].

#### 2.2.2. Specific Setting

Sierra Leone has 34 government hospitals and 1320 peripheral health units [16]. The study was conducted at three selected tertiary hospitals located in Freetown, the capital city of Sierra Leone and the epicenter of the COVID-19 outbreak [17]. These facilities were chosen as most of the healthcare workers had received IPC training and about 42% of Sierra Leone’s healthcare workers are in Freetown [18].

##### Connaught Hospital

The hospital was established in 1912 to provide healthcare services to freed slaves and named in memory of the Duke of Connaught. Today, the hospital is a tertiary government referral center and is supported by the Government of Sierra Leone through the Ministry of Health and Sanitation. The hospital offers a range of medical and surgical services through 25 departments/units and has 16 wards with more than 300 beds [19]. It also has the highest number of specialists in the country with both inpatient and outpatient services and is part of the University of Sierra Leone Teaching Hospital Complex (USLTHC), which was established to support postgraduate training.

##### Princess Christian Maternity Hospital

Princess Christian Maternity Hospital (PCMH) is a tertiary referral government hospital that provides obstetric and gynecological healthcare services. It receives support mainly from the Government of Sierra Leone through the Ministry of Health and Sanitation with the Free Health Care Initiative (FHCI) introduced in 2010 to provide free healthcare services to pregnant women, lactating mothers and under-five children [20]. It has eight wards (six obstetrics wards, one gynecology ward and a high-dependency unit and one labor ward) with a capacity of over 140 beds, and it is part of the USLTHC, which was established to support postgraduate training [21]. 

##### Ola during Children’s Hospital

The Ola During Children’s Hospital (ODCH) is the national referral pediatric hospital. It started as a community self-help project which the government took over and renovated in 1961. The hospital has an inpatient capacity of 139 beds and comprises an emergency room, intensive care units, step down (transition ward), therapeutic feeding center, Special Care Baby Unit (SCBU) or neonatal unit, observation ward, resuscitation ward and two general wards [22]. The hospital is part of the USLTHC, which was established to support pediatric postgraduate training.

### 2.3. Study Population and Period

The study was conducted in PCMH, ODCH and Connaught hospitals, which account for approximately 60% of COVID-19 infections among the healthcare workers (HCWs) in the country. They were in the epicenter (capital city) of the COVID-19 outbreak. These facilities account for 12% of all the healthcare workers in the country [18]. Based on hospital records, in 2021, there were 4500, 15,950 and 8817 inpatient admissions at Connaught, ODCH and PCMH, respectively. In each of these healthcare facilities, there are on average 400 healthcare workers to provide 24-hour healthcare service to clients. 

These facilities were part of the nationwide assessment carried out in 2019 using the IPCAF tool as a pilot project. Additionally, these facilities established isolation units where suspected COVID-19 patients were admitted until COVID-19 laboratory results were released. All the staff working at the isolation units and a small number of staff who provide routine healthcare services to patients had received training on IPC for COVID-19. 

### 2.4. Data Collection, Variables and Analysis

The WHO IPCAF was used to assess IPC programs and activities in these three tertiary hospitals. The tool is a self-assessment tool that should be administered by the facility IPC focal person; however, it can also be used for joint assessment by external assessors. The IPCAF is divided into eight sections, which reflect the eight WHO ‘Core Components of Infection Prevention and Control Programmes’. These are:Core component (CC) 1: IPC program;CC2: IPC guidelines;CC3: IPC education and training;CC4: healthcare-associated infection surveillance (HAI);CC5: multimodal strategies for implementation of IPC interventions;CC6: monitoring/audit of IPC practices and feedback;CC7: workload, staffing and bed occupancy;CC8: environments, materials and equipment for IPC.

For each CC, a maximum score of 100 points can be achieved. Hence, the highest possible overall IPCAF score is 800 points. Depending on the overall score, an IPC grade is allocated to a healthcare facility. Scores from 0 to 200 points correspond to ‘Inadequate’, 201–400 points indicate ‘Basic’, 401–600 points indicate ‘Intermediate’, and 601–800 points indicate ‘Advanced’ IPC compliance (Table 1). IPC performance in each component was graded based on the obtained percentage: (i) ‘Inadequate’ (0–25%), (ii) ‘Basic’ (25.1–50%), (iii) ‘Intermediate’ (50.1–75%) and (iv) ‘Advanced’ (75.1–100%). IPC subcomponents attaining the maximum score were considered ‘strengths’. IPCAF subcomponents with zero or inadequate scores (≤25%) were considered ‘gaps’.

The IPCAF tool was filled in by the IPC focal person of each hospital (who consulted other stakeholders within the hospital to complete the relevant components), and the information was cross-validated by the principal investigator. Additionally, direct observations were carried out by the principal investigator where necessary. 

A Microsoft Excel spreadsheet was created for this study, and the 2021 prospective data were added to the workbook. The principal investigator screened the datasets to check for incompleteness and inconsistencies and ensure data quality. 

A descriptive analysis of each core component of IPC was carried out at the different healthcare facilities followed by a comparative analysis. Average scores for each of the core components and subcomponents were computed for each healthcare facility. 

## 3. Results

### 3.1. IPC Compliance

The overall IPC compliance scores were 333.5 of 800 for Connaught Hospital, 323.5 of 800 for Ola During Children’s Hospital and 296 of 800 for Princess Christian Maternity Hospital. These equate to a ‘Basic’ level of compliance grade at these three tertiary healthcare facilities (Table 2).

For the individual core components, the majority of the scores for the three tertiary healthcare facilities ranged from ‘Inadequate’ to ‘Basic’. The components with the least scores were ‘healthcare-associated infection surveillance’. The component with the best level of compliance was the ‘IPC programme’. Additionally, Connaught Hospital scored higher in ‘built environment, materials and equipment’ as compared to the other two healthcare facilities (Table 3). Further details on the different component scores are shown in Appendix A. 

### 3.2. Strengths and Gaps

The major strengths and gaps at the three tertiary healthcare facilities related to the different components of the IPC framework are shown in Table 4 below. The strengths and gaps at these healthcare facilities were similar. There were gaps in all of the components, but the most numerous were observed for the following components of the IPCAF tool: IPC guidelines, HAI surveillance, monitoring/audit of IPC practices and built environment, materials and equipment.

#### 3.2.1. IPC Program, Guidelines and Education and Training

The three tertiary healthcare facilities had an IPC program with a dedicated IPC focal person. However, there was no dedicated IPC budget to support the implementation of activities. The IPC focal persons at Connaught and Ola During Children’s hospitals had received the national advanced IPC training, while the IPC focal person at Princess Christian Maternity Hospital had yet to receive any formal training in IPC. Both Connaught and Ola During Children’s hospitals had access to a microbiology laboratory onsite, whereas Princess Christian Maternity hospital accessed the ODCH microbiology laboratory. However, they were not reporting results of ‘culture and sensitivity’ testing.

#### 3.2.2. Healthcare-Associated Infection Surveillance, Multimodal Strategy and Monitoring/Audit of IPC Practices

The three healthcare facilities used the national IPC guidelines (2015), which did not include guidance on outbreak management and preparedness, prevention of HAI or prevention of transmission of multidrug-resistant pathogens. There was no well-defined HAI surveillance protocol, including surveillance of infections that affect healthcare workers in the clinical, laboratory and other settings. Even though a multimodal strategy was employed for IPC intervention, a multidisciplinary approach was not being used. There were no clearly defined objectives and targets to monitor IPC activities. The healthcare facilities monitored hand hygiene compliance quarterly and conducted the WHO Hand Hygiene Self-Assessment Framework Survey annually. 

#### 3.2.3. Workload Staffing and Bed Occupancy and Built Environment Materials and Equipment

Among the healthcare facilities, there was no agreed healthcare worker-to-patient ratio, and staffing levels were not assessed according to patient load. Single patient rooms for grouping patients with similar pathogens were not available. Each facility had a burning pit/waste dump, incinerators were non-functional, and the areas for cleaning, disinfecting and sterilizing medical devices/equipment were not functioning reliably. Personal protective equipment such as examination gloves, facemasks and aprons were not continuously available and had been largely donated to these healthcare facilities.

Despite these similarities, there were some marked differences. At Connaught Hospital, water services were available and of sufficient quantity while at Ola During Children’s Hospital and Princess Christian Maternity Hospital, water supply was available on average less than five days per week. The water supply in all facilities was not considered safe for drinking by staff, patients or family members. At the Ola During Children’s Hospital, there was sufficient power supply present at all times and in all departments while at Connaught and Princess Christian Maternity Hospitals there was insufficient power supply in all areas, day and night. Connaught and Ola During Children’s hospitals had the required number of functional toilets in their facilities, while this was not the case at Princess Christian Maternity Hospital.

## 4. Discussion

This is the first study conducted in tertiary hospitals in Sierra Leone using a globally accepted and standardized tool such as the WHO IPCAF tool to evaluate the effectiveness of implementation of IPC programs at healthcare facilities. All of the three study hospitals scored a ‘Basic’ grade for IPC implementation. This implies suboptimal implementation, indicating a great scope for improvement. These findings are similar to those found in a study carried out at the Lira university hospital in Uganda [23]. In contrast to our findings, a study conducted in five public hospitals in Islamabad, Pakistan, had all scored ‘Inadequate’ [24]. Of the eight components of the IPC programs, it was observed that HAI surveillance was ‘Inadequate’, six components were ‘Basic’, and only the IPC program scored ‘Intermediate’. This suggests that none of the eight core components were effectively implemented at the three healthcare facilities. This might be related to lack of a dedicated budget, suboptimal education and training, no surveillance and monitoring, weak infrastructure and insufficient materials and equipment.

A positive finding was that all three study hospitals had an IPC program, an IPC committee and a dedicated IPC focal person. However, the IPC committees were not multidisciplinary and did not provide oversight of the IPC focal person in the implementation of IPC interventions at their facilities. These findings are consistent with studies from Georgia and Tanzania [25,26]. A key challenge was lack of a dedicated budget in the three study sites. This is also true for the national IPC program in Sierra Leone and many IPC programs in sub-Saharan Africa [27]. 

An uninterrupted supply of clean and safe water has been a challenge in most healthcare facilities in Sierra Leone. Of the three facilities in our study, only one facility had a water supply seven days a week. None of the facilities had constant safe drinking water for patients, visitors or healthcare workers. These findings are similar to those of a study in Pakistan [24]. The challenge for clean and safe water is seen in many low- and middle-income countries. The picture in high-income countries is completely different, as studies conducted in Austria, Georgia and Germany revealed that the majority of the healthcare facilities had uninterrupted running water and electricity supply at all times [25,28,29]. There is evidence that healthcare workers working in facilities with a constant water supply in their department are 1.6 times more likely to have good Infection Prevention and Control practices as compared to HCWs working in facilities without a continuous water supply in their departments [30]. 

IPC supplies such as hand gloves, face masks, and aprons were not readily available in any of the three healthcare facilities. This is in line with findings from Ghana, wherein only 19 out of the 56 healthcare facilities had a sufficient quantity of examination gloves, face masks, aprons and other personal protective equipment [31]. To promote effective and standard clinical practice in accordance with guidelines, emphasis should be placed on optimizing the healthcare environment and the availability of IPC supplies. A WHO expert panel recommends that materials and equipment for performing appropriate IPC measures such as hand hygiene should be readily available at the point of care [32].

Our study revealed that the hospital IPC focal persons have the ability and capacity to monitor IPC practices and feedback. However, there was no defined monitoring plan with clear goals, targets and activities at these three healthcare facilities. The national IPC officers are implementing the WHO hand hygiene self-assessment framework at these healthcare facilities annually as recommended by WHO. Implementing the hand hygiene survey is a good practice as it is a key indicator for evaluating the implementation of an IPC program at healthcare facilities [33]. Our findings are consistent with a similar study conducted in Georgia where only 7 out of 41 healthcare facilities had an IPC monitoring/audit plan; however, none of these plans had all the necessary elements, such as clear goals and objectives, tools to systematically collect data, clearly defined roles and responsibilities and a work plan or schedule [25]. In Sierra Leone, several monitoring tools have been developed or adapted by the national IPC unit to support hospital IPC focal persons in conducting routine assessments at their facilities. These include the national IPC/WASH tool, IPC scorecard and hand hygiene observation tool. However, these assessments are not performed regularly enough by the IPC focal persons to identify the gaps and challenges associated with the implementation of the IPC program, and the resulting findings are not used to guide the implementation of activities. We recommend consistent use of the WHO IPCAF tool at all secondary and tertiary healthcare facilities to monitor the trend and progress of the implementation of the IPC program and suggest interventions that will improve IPC program implementation.

A WHO panel of experts recommends that facility-based HAI surveillance should be performed to guide IPC interventions and detect outbreaks, including AMR surveillance with timely feedback of results to healthcare workers and stakeholders [32]. None of the three tertiary healthcare facilities were conducting routine HAI surveillance for surgical site infections, catheter-associated urinary tract infections, central-line-associated bloodstream infections or ventilator-associated pneumonia during the study period. However, ad hoc surgical site infection surveillance activities are conducted by researchers. In contrast to our findings, a recent study from Germany which assessed 736 hospitals using the IPCAF tool confirmed that HAI surveillance, as well as monitoring and audit of IPC practices, were well-established [29]. The three healthcare facilities in Sierra Leone were not conducting routine surveillance activities due to the following reasons: weak bacteriology capacity, poor information and communication support, limited expertise in conducting surveillance activities and no budget to implement HAI surveillance. It is recommended that the bacteriology capacity should be improved at these facilities. Additionally, the national IPC program, with support from partners, should develop and implement a national HAI surveillance strategy that can be used at the healthcare facilities. This will support healthcare facilities in evaluating the burden of HAI to ascertain the national HAI burden. There is a light at the end of the tunnel as these three healthcare facilities may start to report on HAI surveillance data by the second or third quarter of the year 2022 as a result of the implementation of the Fleming fund grant system to improve bacteriology capacity at these tertiary healthcare facilities. 

Our study had several strengths. First, we addressed identified national and global operational research priorities. Second, data collection was carried out by IPC focal persons and validated by the principal investigator, all of whom were well-versed in IPC terminologies. Third, we used a structured and validated data collection pro forma, the WHO IPCAF tool. This facilitated implementation of uniform procedures in data collection. Fourth, we adhered to ‘STROBE’ (Strengthening the Reporting of Observational Studies in Epidemiology) guidelines for data collection and reporting of study findings.

There were some limitations to our study. First, this was a one-time study and hence provides only a baseline assessment. Follow-up assessments should be conducted to track progress. Second, our findings are not generalizable as only three healthcare facilities located in the capital city were included in the study. A wider study using a representative sample of health facilities across the country is needed.

Finally, this was a self-assessment by the hospital IPC focal persons, which might have led to overestimation of scores in certain components. However, the level of bias was reduced as the principal investigator cross-validated the responses, and the tool was developed by WHO as a self-assessment instrument.

### 4.1. Recommendations

#### 4.1.1. Low Cost

The hospital IPC teams, with both technical and financial support from the national IPC unit, WHO, Centres for Disease Control (CDC) and other agencies, should conduct new employee orientation and training for all healthcare workers and administrative staff. Additionally, continuous professional development programs should be made available to all IPC focal persons to improve their knowledge and understanding of IPC.

#### 4.1.2. Medium Cost

The national IPC unit and its implementing partners should develop a national HAI surveillance strategy and support the hospital IPC team in conducting regular HAI surveillance. The national IPC unit and hospital IPC teams should conduct quarterly implementation of the WHO IPCAF tool at healthcare facilities to monitor the implementation of IPC programs at national and facility levels. IPC materials, such as examination gloves, face masks, aprons and other IPC materials, should be supplied in an uninterrupted manner to protect healthcare workers and patients from HAI and AMR.

#### 4.1.3. High Cost

The government of Sierra Leone, through the Ministry of Health and Sanitation and its implementing partners, should provide technical and financial support (especially a dedicated budget for IPC) to the national and hospital IPC team for the implementation of the IPC program at healthcare facilities to reduce the burden of HAI and AMR.

## 5. Conclusions

Our study found that all of the three tertiary healthcare facilities in Freetown scored a ‘Basic’ level of IPC compliance, and several gaps were identified in all eight WHO core components, specifically in built environments, materials and equipment, monitoring/audit of IPC practices and HAI surveillance. Availability of a clean, safe and uninterrupted water supply is essential for the implementation of the IPC program at the facility level, and the purchase of IPC supplies, such as personal protective equipment, hand hygiene stations and cleaning agents, should be a priority. Furthermore, a dedicated IPC budget is necessary for the implementation of IPC activities. As the country has updated its national IPC guidelines to reflect the WHO core component approach, we recommend that healthcare workers be trained on the updated national guidelines, emphasizing HAI surveillance and monitoring/audit of IPC practices. All these activities have the potential to reduce the burden of HAI and AMR on our healthcare facilities and support quality healthcare service delivery and the achievement of universal health coverage by the year 2030. Our study findings have added to the national and global body of evidence on the implementation of IPC programs at healthcare facilities.

## Figures and Tables

**Table 1 ijerph-19-05275-t001:** The WHO IPCAF tool grading and interpretations.

Score	Grading	Interpretation
0–200	Inadequate	Implementation of IPC core components is deficient. Significant improvement is required
201–400	Basic	Some aspects of the IPC core components are in place but not sufficiently implemented. Further improvement is required
401–600	Intermediate	Most aspects of the IPC core components are appropriately implemented. The facility should continue to improve the scope and quality of implementation and focus on the development of long-term plans to sustain and further promote the existing IPC program activities
601–800	Advanced	The IPC core components are fully implemented according to the WHO recommendations and appropriate to the needs of the facility

WHO—World Health Organization. IPCAF—Infection Prevention and Control Assessment Framework.

**Table 2 ijerph-19-05275-t002:** Baseline level of IPC compliance as measured by the IPCAF tool at three tertiary healthcare facilities in Freetown, Sierra Leone, 2021.

Healthcare Facility Name	IPCAF Score (Max = 800)	Interpretation
Connaught Hospital	333.5	Basic
Ola During Children’s Hospital	323.5	Basic
Princess Christian Maternity Hospital	296	Basic

IPC—Infection Prevention Control; IPCAF—Infection Prevention and Control Assessment Framework at facility level. Maximum IPCAF score was 800: 0–200 Inadequate; 201–400 Basic; 401–600 Intermediate; and 601–800 Advanced.

**Table 3 ijerph-19-05275-t003:** Baseline level of compliance for each core component of the IPC programs at three tertiary healthcare facilities in Freetown, Sierra Leone, 2021.

Core Components	Score Interpretation		
	Connaught	ODCH	PCMH
IPC program	Intermediate	Intermediate	Intermediate
IPC guideline	Basic	Basic	Basic
IPC education and training	Basic	Basic	Basic
HAI surveillance	Inadequate	Inadequate	Inadequate
Multimodal strategies	Basic	Basic	Basic
Monitoring/audit of IPC practice	Basic	Basic	Basic
Workload, staffing and bed occupancy	Basic	Basic	Basic
Built environment, materials and equipment	Intermediate	Basic	Basic
Overall score	Basic	Basic	Basic

Maximum score for each component was 100. Core component score interpretations: 0–25% Inadequate; 25.1–50% Basic; 50.1–75% Intermediate; and 75.1–100% Advanced. IPC = infection prevention control; HAI = healthcare-associated infections.

**Table 4 ijerph-19-05275-t004:** Strengths and gaps in the different core components of the IPC program at the Connaught, Princess Christian Maternity and Ola During Children’s hospitals in Freetown, Sierra Leone, 2021.

Core Components	Strengths	Gaps
IPC program	Each facility has an IPC program Dedicated IPC focal person The facilities have access to a microbiology laboratory	No dedicated budget for the IPC program
IPC guideline	Each facility has standard precautions, hand hygiene, transmission-based precautions, disinfection and sterilization, healthcare worker protection and safety, injection safety and waste management guidelines	No written guidelines for: Outbreak management and preparedness; Prevention of vascular-catheter-associated bloodstream infections; Prevention of all types of hospital-acquired pneumonia; Prevention of catheter-associated urinary tract infections; Prevention of transmission of multidrug-resistant (MDR) pathogens.
IPC education and training	IPC focal person has completed the national IPC advanced training and has the expertise to lead IPC training except at Princess Christian Maternity Hospital. Non-IPC personnel with adequate skills to act as trainers and mentors except at Ola During Children’s Hospital.	Regular IPC trainings are not conducted for healthcare workers and administrative staff IPC training is not yet integrated in clinical practice and training of specialists No IPC training for patients or family members to minimize healthcare-associated infections No certified continuous professional development courses for IPC focal person. However, they attend refresher trainings
HAI surveillance	HAI surveillance is a defined component of each facility IPC program There is a person responsible for surveillance activities	No information technology support to conduct surveillance activities Facilities are not conducting surveillance of surgical site infections and device-associated infections (catheter-associated urinary tract infection, central-line-associated bloodstream infection and ventilator-associated pneumonia). Only Princess Christian Maternity Hospital was found to be conducting SSI surveillance. No surveillance of infections that may affect healthcare workers in the clinical, laboratory or other settings No analysis of antimicrobial drug resistance data due to lack of microbiology capacity
Multimodal strategies	Each facility uses a multimodal strategy to implement the IPC program. These strategies include the use of different tools or checklist IPC focal person links with colleagues at the quality improvement unit to develop and promote IPC multimodal strategies	Safety climate and culture change is not included in the multimodal strategy A multidisciplinary team not used to implement multimodal strategy
Monitoring/audit of IPC practice	IPC focal person has the ability and capacity to monitor IPC practice and provide feedback across all facilities Each facility monitors hand hygiene compliance WHO Hand Hygiene Self-Assessment Framework Survey has been undertaken annually by each facility The state of the IPC activities and compliance are reported to department leaders, managers and frontline healthcare workers	No defined monitoring plan with clear goals, targets and activities No facilities monitor: Intravascular catheter insertion and/or care; Wound dressing drainage; Consumption of alcohol-based hand rub.
Workload, staffing and bed occupancy	Bed occupancy is maintained at one patient per bed across all the facilities	Staff level not assessed according to patient workload No agreed healthcare worker-to-patient ratio across the facilities No system in place to assess and respond when bed capacity is exceeded There is inadequate bed spacing in certain departments across all the facilities
Built environment, materials and equipment	Functioning hand hygiene stations present at all points of care There is natural environmental ventilation in patient care areas Wastewater treatment system present and functioning	No reliable safe drinking water available for staff, patients and family members at all times and in all locations No single patient rooms for grouping patients with similar pathogens The constructed burning pit/waste dump in the facilities has insufficient dimensions Non-functional incinerator in the facilities Disposable items, such as examination gloves, facemasks and aprons, are not continuously available

IPC = infection prevention control; HAI = healthcare-associated infections; SSI = surgical site infection.

## Data Availability

The dataset used in this paper has been deposited at https://doi.org/10.6084/m9.figshare.19188170 (accessed on 26 January 2022) and is available under a CC BY 4.0 license.

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
