# Peer review of "Infection Prevention and Control in Three Tertiary Healthcare Facilities in Freetown, Sierra Leone during the COVID-19 Pandemic: More Needs to Be Done!"

_ijerph, 2022, doi:10.3390/ijerph19095275_

Round 1
Reviewer 1 Report
Comments to the Article “Infection prevention and control in three tertiary healthcare facilities in Freetown, Sierra Leone during the COVID-19 Pandemic: More needs to be done!”, by Ibrahim Franklyn Kamara et al.
The authors in this article report findings on a hospital-based cross-sectional study developed to assess the Infection Prevention and Control compliance at health care facilities in Sierra Leone’s capital.
The article contains expected components (Introduction, Methods, Results, Discussion and Conclusion). The Introduction section is concise yet properly introduces the problem in analysis. The Methods section, as well as the Results and Discussion sections, are exhaustive and illustrative.
The comparison of findings with results from other studies is mentioned.
Strengths and limitations of the study are also mentioned.
The work presented in the article is no more than a descriptive report of scores for each of the three facilities considered.
Author Response
Comment: The work presented in the article is no more than a descriptive report of scores for each of the three facilities considered.
Response: Thank you very much for your comments.
Reviewer 2 Report
In this study ‘Infection prevention and control in three tertiary healthcare facilities in Freetown, Sierra Leone during the COVID-19 Pandemic: More needs to be done!’ the authors have performed a hospital-based cross-sectional study in three different at three Connaught hospitals, Princess Christian Maternity hospital, and Ola During Children’s hospital located in Freetown, Sierra Leone. They have used the World Health Organization’s Infection Prevention and Control Assessment Framework Tool to assess the level of IPC compliance at these healthcare facilities. I would encourage the author(s) to carry out the following revisions – which should potentially present a better case and improve the manuscript.
Table 3 describes the Baseline level of compliance for each core component of the IPC programs at three tertiary healthcare facilities. I would suggest the authors to make the table more intriguing by providing the individual value for giving the final score (Core component score interpretations: 0-25% Inade-184 quate; 25.1-50% Basic; 50.1-75% Intermediate; and 75.1-100% Advanced). For example, what are the IPC practices monitored (hand hygiene, MDR pathogen screening, disinfection, sterilization, etc.)? and how each of these factors was scored under IPC practices?
Author Response
Comment: Table 3 describes the Baseline level of compliance for each core component of the IPC programs at three tertiary healthcare facilities. I would suggest the authors to make the table more intriguing by providing the individual value for giving the final score (Core component score interpretations: 0-25% Inadequate; 25.1-50% Basic; 50.1-75% Intermediate; and 75.1-100% Advanced). For example, what are the IPC practices monitored (hand hygiene, MDR pathogen screening, disinfection, sterilization, etc.)? and how each of these factors was scored under IPC practices?
Response: Thank you very much for your comment. We have modified the table as suggested and have added this as a supplementary table – this will help in keeping the table 3 concise and the details are available in the supplementary table.
Reviewer 3 Report
The analyzed manuscript addresses a very important problem of the prevention of nosocomial infections and antibiotic resistance.
It is a retrospective survey of 3 hospitals in Sierra Leone.
1) The weakness of this study was the survey itself, as if I understand correctly the results indicated by the designated person in the analyzed hospitals were not controlled by the authors of the studies. Please clarify whether the given results of the IPCAF questionnaire were verified by the authors of the paper as to the reliability of the data provided by the hospitals.
2) In line 53, please update your literature with the latest data on the efficacy of remdesivir, molnupiravir and nirmatrelvir / ritonavir, aren't really any of them effective in treating COVID-19?
3) Please clarify what constituted the choice of the analyzed group, 3 hospitals in the capital of Sierra Leone were analyzed, although their selection was not entirely clear to me.
4) In assessing which hospital has been selected, please indicate how many of these hospitals have beds compared to the rest of the country, how many hospitalizes patients annually, and what the average rate of nosocomial infections they have, and also, if this data is available, how many percents of healthcare workers received IPC training in different hospitals?
5) It is indicated that 42% of healthcare workers work in the capital, please specify how many of them work in designated hospitals, can this be considered a representative percentage for the rest of the country?
6) The study is very interesting without linking it to the ongoing COVID-19 pandemic, I was not able to fully follow the authors' thought linking the ongoing pandemic with the analyzed microbiological and sanitary situation in the analyzed hospitals. It is indicated that the analyzed hospitals had designated places for COVID-19 patients, and that infections were observed among staff, I would prefer not to associate this study with a pandemic, or if so to indicate what percentage of COVID-19 patients were hospitalized in different hospitals, how many healthcare workers cought COVID-19. What was the impact of IPC on COVID-19 infections.
In my opinion, this work does not require any links to an ongoing pandemic and is interesting in itself from a microbiological point of view.
Author Response
Comment 1: The weakness of this study was the survey itself, as if I understand correctly the results indicated by the designated person in the analyzed hospitals were not controlled by the authors of the studies. Please clarify whether the given results of the IPCAF questionnaire were verified by the authors of the paper as to the reliability of the data provided by the hospitals.
Response: Thank you very much for your comment. The tool is developed as a self-assessment tool by WHO. Line 181-184, we stated clearly that the responses completed by the IPC focal were cross-validated and direct observations were made when necessary by the principal investigator.
Comment 2: In line 53, please update your literature with the latest data on the efficacy of remdesivir, molnupiravir and nirmatrelvir / ritonavir, aren't really any of them effective in treating COVID-19?
Response: Thank you very much for your comment. We have rephrased the statement. We have refrained from providing any latest data on efficacy of antiretrovirals as that is not the topic of the paper.
Comment 3: Please clarify what constituted the choice of the analyzed group, 3 hospitals in the capital of Sierra Leone were analyzed, although their selection was not entirely clear to me.
Response: Thank you very much for your comment. We have elaborated on the reasons for selecting these three hospitals in lines 137-147.
Comment 4: In assessing which hospital has been selected, please indicate how many of these hospitals have beds compared to the rest of the country, how many hospitalizes patients annually, and what the average rate of nosocomial infections they have, and also, if this data is available, how many percents of healthcare workers received IPC training in different hospitals?
Response: Thank you very much for the comment. We have added the total number of inpatients admission in 2021. We have highlighted that all the workers at the isolation units and some at other departments have received training on IPC for COVID-19. Data is not available for the average nosocomial infections.
Comment 5: It is indicated that 42% of healthcare workers work in the capital, please specify how many of them work in designated hospitals, can this be considered a representative percentage for the rest of the country?
Response: Thank you very much for the comment. We have included the percentage working at the three healthcare facilities in line 139-140.
Comment 6: The study is very interesting without linking it to the ongoing COVID-19 pandemic, I was not able to fully follow the authors' thought linking the ongoing pandemic with the analyzed microbiological and sanitary situation in the analyzed hospitals. It is indicated that the analyzed hospitals had designated places for COVID-19 patients, and that infections were observed among staff, I would prefer not to associate this study with a pandemic, or if so to indicate what percentage of COVID-19 patients were hospitalized in different hospitals, how many healthcare workers cought COVID-19. What was the impact of IPC on COVID-19 infections? In my opinion, this work does not require any links to an ongoing pandemic and is interesting in itself from a microbiological point of view.
Response: Thank you very much for your comments. We have improved on the introduction and methods sections to link the study to the ongoing COVID-19 pandemic. There is high rate of healthcare worker infection in SSA including Sierra Leone and an effective IPC program in the healthcare facilities will reduce the burden of HAI.
Round 2
Reviewer 2 Report
The manuscript can be accepted in the present form, however I found these two references [1. Squire, J. S.; Dadzie, D.; Nyarko, K. M.; Danso-Appiah, A.; Kaburi, B. B.; Noora, C. L.; Vandi, M. A.; Ameme, D. K.; Kenu, E.; Sackey, S. O., Risk Factors for COVID-19 infection among Hospital Healthcare Workers, Sierra Leone, 2020. Journal of Interventional Epidemiology and Public Health 2022, 5, (4). 2. Organization, W. H. Human Resources for Health Country Profile; 2016.]. at the end of the reference section, in page 14. Both of these references need to be properly inserted in the text and in the reference [ There are already two references as 1 and 2 in page 12].
Author Response
Thank you very much for your comment.
The references and intext citations have been done accurately.
Reviewer 3 Report
The corrections made fill my comments on the manuscript. I believe that the article should be accepted for publication in this form.
Author Response
Thank you very much for your comment.